# Using Simulations to Help Public Health Students Overcome Language Barriers for Better Health Outcomes

**DOI:** 10.3390/ijerph20136259

**Published:** 2023-06-30

**Authors:** Hilde Skjerve, Lars Erik Braaum, Ursula Småland Goth, Anette Sørensen

**Affiliations:** 1School of Health Sciences, Kristiania University College, Kirkegata 24-26, 0152 Oslo, Norway; larserik.braaum@kristiania.no (L.E.B.); anette.sorensen@kristiania.no (A.S.); 2NLA University College, Campus Oslo, P.B 7153 St. Olavs Plass, 0130 Oslo, Norway; ursula.goth@nla.no

**Keywords:** health outcomes, communication, education, health, migration, qualitative methods, simulation, training

## Abstract

Growing migration into Norway has increasingly strained the country’s health services. Good communication is essential to ensure quality care. Often, healthcare workers and immigrant clients do not share a common language, and it is known that the conditions and expectations of immigrant clients can be different from the majority population. This study aimed to explore the viability of utilizing simulations as a pedagogical tool for educating public health students in effectively navigating a multicultural environment to promote better health outcomes. This study is a component of an extra-curricular training project that utilized a convergent mixed-methods design. The present study focuses on reporting the qualitative component of the findings. The data collection process encompassed the implementation of a stepwise simulation exercise with case-based clinical scenarios focusing on three lifestyle diseases specifically designed for this study. Method triangulation was achieved by using different methodological approaches in the analysis. Our results show the importance of simulation training for healthcare students when working with clients who do not share the same language. Interactions with clients of different backgrounds must be practiced, and simulations can be used to improve healthcare students’ communication skills. The study highlights the need for healthcare education programs to integrate cultural competence simulation training and broaden the scope of medical training to address culturally challenging encounters.

## 1. Introduction

There has been a great increase in cultural diversity in Norway owing to the many ethnic groups now living and working in the country. In recent decades, Norway can be said to have become a multilingual and multicultural society. Today, Norway consists of several groups that consider themselves, and are recognized by others, as culturally different from each other. Few Norwegians have experience with these “other people’s” cultures or languages [1,2]. Yet Norway still asks its health workers to provide equal care to everyone, regardless of background [1,3,4]

Today, health professionals encounter not only language barriers and cultural differences, but also special health challenges from some groups. For many immigrants from non-western countries, the Norwegian understanding of illness can seem unfamiliar and even threatening. This distinction can create challenges when providing assistance to immigrant individuals [5]. Therefore, health workers must have good communication skills along with awareness of the many cultural challenges [6]. Such knowledge is best acquired during their academic education [7]. Nurses, therefore, are requesting practical teaching about communication, immigration, and health [3]. In 2010–2011, the Norwegian government announced that everybody will have equal healthcare services, and by 1 January 2022, the Interpretation Act was ratified for the public sector [8]. The way healthcare professionals interact with individuals from diverse ethnic backgrounds will ultimately determine the true equality of the service provided.

Skill training through simulations has proven to be a learning method that prepares one for the real world after graduation. There are several forms of simulation, from simple skills training at the individual level to complex interactions and decision training with teams. It has been shown that newly qualified nurses have low confidence in the face of demanding and complex client situations [9], and thus experience a high degree of stress related to the demands of the job [10,11]. Both students and teaching staff, therefore, have called for more practical teaching that can help them master complex situations [12]. Simulation, for practical purposes, can be defined as an instructional method employed by teachers to recreate or mimic real-life events, problems, procedures, or skills with the aim of achieving specific educational objectives [13]. In simulated environments, students are assigned roles or tasked with completing specific activities. The ultimate goal is to provide students with a realistic perspective, allowing them to apply and practice new skills and knowledge, think critically, and derive meaning from the scenario [14]. Such didactic approaches will increase the understanding of theory through practice of procedures. Regular training in complex situations and structured communication will increase the quality of client care. Therefore, quality-enhancing measures that are developed with simulation-based training will ensure satisfactory client care among healthcare professionals [15,16].

### Using Simulations to Train Healthcare Students

Modern health services involve individuals working together in teams. Client care requires that each team member masters advanced treatment and care in acute and complex situations [17]. Healthcare simulation is the expert use of established simulation methodology and techniques, requiring individual judgment and a holistic understanding of that method and its context [18]. During a simulation, the team is given the opportunity to practice communication skills with the aim of creating a common understanding of the client’s situation. The debriefing after the simulation stimulates critical reflection of each member’s and the group’s performance [19]. As an instructional approach, simulation has the potential to enhance learning by incorporating the human factor, where individuals’ experiences and perspectives can influence outcomes within the classroom setting [14].

Adverse events are injuries or maltreatment due to inappropriate medical care and represent a major source of morbidity and mortality worldwide. Therefore, the WHO outlined the future direction of client safety research focusing on challenges with communication [20]. A recent Norwegian study showed that immigrants still experience difficulties in understanding and being understood because of linguistic and cultural differences [21]. Therefore, we aimed to investigate a didactic approach to avoid impediments that some migrants experience when accessing healthcare services. We could not find any studies focusing on simulations in teaching communication challenges in public health. This is the first study investigating this field.

Research question: How does using simulations as a didactic approach influence the language and communication challenges experienced by healthcare students and individuals with immigrant backgrounds, and how does it potentially enhance health workers’ interaction skills?

## 2. Materials and Methods

This study is a component of an extra-curricular training project that utilized a convergent mixed-methods design. The present study focuses on reporting the qualitative component of the findings. The decision to solely present the qualitative portion of the results was motivated by the research question’s specific emphasis on exploring participants’ experiences regarding communication challenges, as well as their perceptions and perspectives about simulation training and its role in enhancing the interaction skills of public health students when communicating with clients who did not share a common language.

The project was designed and implemented as a stepwise simulation exercise in the spring of 2022 at Kristiania University College. Student teams were assigned the task of navigating the scenarios, which revolved around interactions with immigrant patients. Throughout the exercise, students were required to adapt their communication strategies to bridge the language barrier and ensure effective information exchange. By engaging in these simulations, students had the opportunity to develop and refine their skills in cross-cultural communication, empathy, and patient-centered care.

### 2.1. Sample

Purposive sampling was used with the following inclusion criteria: second- and third-year bachelor students in a public health program at Kristiania University College who had completed three semesters with 90 ETCS and the Motivational Communication class. A total of 54 students received an invitation through Canvas—a learning management system—to join the study. Initially, twenty students provided their informed consent to participate in the study. However, two students had to withdraw due to health reasons, resulting in the participation of only eighteen students. The interviews were not electronically recorded; the facilitators manually recorded all data into logbooks. Students completed evaluation forms as part of the process.

### 2.2. Data Collection

The data emerged from questionnaires, logbooks from four facilitators, and evaluation forms filled out by the students. The questions were copied from questionnaires that are used for training at Norwegian hospitals. The students’ reflections and quotations were noted in logbooks (A–F). Evaluation forms that contained the participants’ reflections were based on twelve open-ended questions that were given after the end of the second simulation review. All facilitators kept the questionnaires before, during, and after the simulation.

### 2.3. Literature Search

During the spring of 2022, a literature search was conducted to gather relevant background information, provide contextual understanding for our research, refine our research question, and inform our methodological approach. Oria, a search engine that helps explore print and electronic collections at the Kristiania University College Library, as well as other university and college libraries in Norway, was used with the keywords “simulation AND health service AND immigrant* AND training”. Additionally, we included the keywords “equivalent health services OR cultural competency AND education OR university training”. The search was limited to the Norwegian, Swedish, Danish, English, and German languages for the years 2012 to 2022. Thirty-nine scientific publications were found, of which ten were relevant to the study and its Norwegian context. These ten articles were included in this study to develop data collection tools. The purpose of this review was to gather relevant background information and provide contextual understanding for our research.

### 2.4. Data Collection Process

Data on student performance and feedback were collected to assess the effectiveness of the training project. These evaluations, combined with student reflections and observations from facilitators, contributed to the ongoing refinement of the simulation exercise and future improvements in the curriculum. The facilitators carried out one pilot test before the study commenced. Before starting the simulation, a briefing on the situation was given by one of the facilitators. Each simulation consisted of two rounds performed by the same two students together with a live evaluator. Each round lasted approximately 15 min. After the first round, the students were debriefed by the facilitators, then given a short break. Then, the second round was carried out with the same scenario as the first round, wherein reflections and experiences from the first round were used and tested. The entire simulation plus debriefing lasted about 45 min. The experiment took place in a classroom, a location that was already familiar to the students.

To create a realistic setting for the interaction between health workers and immigrants with undisclosed health needs, three live evaluators from Kristiania University College, who were unfamiliar to the participating students, assumed the roles of hospital patients. Among them, one represented an immigrant from a Middle Eastern country, another hailed from Asia, and the third originated from South America. Throughout the scenario, they communicated exclusively in their native languages, which were unfamiliar to the students. Moreover, they feigned having only basic proficiency in Norwegian and English. Each live evaluator was assigned a unique case, featuring a different lifestyle disease. In this manner, the scenario effectively mirrored the challenges that health workers face when engaging with immigrants who possess significant, yet undisclosed, health concerns.

### 2.5. Simulation Step-by-Step Description

The data collection process encompassed the implementation of a stepwise simulation exercise, specifically designed for this study. In this exercise, student teams were allocated hypothetical clinical scenarios involving immigrant patients. These clinical scenarios were case based, focusing on three lifestyle diseases, and presented to the students in two rounds by role players. A debriefing session was conducted between the rounds to facilitate reflection and discussion. This structured approach allowed for the systematic exploration and analysis of the students’ performance and interactions within the simulated environment. The steps of the simulation can be summarized as follows:Briefing: Each simulation commenced with a briefing provided by one of the facilitators, outlining the scenario details.Round 1: In this phase, students interacted with the scenario for approximately 15 min. They engaged with a ‘patient’, portrayed by an actor who was familiar with the simulation but communicated solely in an unfamiliar language to the students.Debriefing and Break: After the first round, students underwent a debriefing session facilitated by the research team. A short break was provided to allow students time for reflection.Round 2: Following the break, students re-engaged with the same scenario, applying insights and experiences gained from the initial round. This round also lasted approximately 15 min.Final Debriefing: After both rounds, a comprehensive debriefing session was conducted to encourage critical reflection on individual and group performance [17].Observation: The principal researcher was present throughout the simulations, actively observing the proceedings and taking visible notes. The potential influence of the Hawthorne effect, where the researcher’s presence may impact the observed processes, was acknowledged and addressed. To mitigate this influence, the experiment was conducted with two different teams of students.

### 2.6. Data Analysis

Method triangulation was obtained by using different methodological approaches (data from the literature search, questionnaires, and observations) included in the analysis. These data were condensed [22] and then categorized through thematic analysis [23] by the authors (HS and USG) and later compared with each other [22,24]. The data were then discussed with the co-authors (LEB and AS). To develop a thorough comprehension of the data, we engaged in an extensive process of reading and re-reading the collected materials. The categories that were formed during the analyses were Communication (experienced challenges) and Simulation as a didactic approach. Through this analysis, meaningful themes relating to communication were identified, taking into account similarities and connections between these themes. The identified themes were meticulously reviewed and refined to ensure their accurate representation of the data and alignment with the research objectives. The main findings that emerged in the analysis were visualized using quotations, providing concrete support and enhancing the credibility of the results, which were reproduced without identification to safeguard the students’ and facilitators’ anonymity [24,25].

### 2.7. Ethical Considerations

The Norwegian Center for Research Data was presented with the study and its methodological approach. Given the study’s design, it was determined that no ethical approvals were required to initiate the research. The study application underwent evaluation by a committee led by a professor of pedagogy at Kristiania University College. The committee consisted of an external pedagogue; the head of the center for learning technology; the director of program innovation, learning technology, and educational development; the head of the center for educational development; and a master’s student. On the 6th of November 2019, prior to commencing data gathering, approval was obtained from the university. However, due to the pandemic, the project’s start was postponed until 2022. The participants were provided with information regarding their rights and the study’s purpose. Verbal and written consent were obtained from all participants.

## 3. Results

The results were categorized into two primary themes: Communication, focusing on experienced challenges, and Simulation as a didactic approach.

### 3.1. Challenges with Communication

The findings of the study revealed several communication challenges experienced by the students during the simulation.

### 3.2. Lack of a Common Language

The findings of this study revealed that both students and role players encountered challenges in communication due to the lack of a common language. All eighteen students expressed difficulty in understanding the immigrant individual, emphasizing the importance of assessing the individual’s level of comprehension. Notably, *“several students highlighted the need for training in communicating with minority individuals as a skill”* (student team 5, as well as students 10, 12, and 15). 

During the first session, facilitator 1 observed that *“once the students realized the client had limited proficiency in Norwegian, they became uncertain and hesitant. They sought advice from one another on how to proceed, leading to a state of silence and uncertainty”*. Facilitator 3 noted that, while *“several groups forgot to inquire about the client’s mother tongue, all of them inquired about the client’s ability to communicate in English”*. It appears that the students were indeed faced with significant communication barriers throughout the simulation. Their struggles to comprehend and effectively interact with the immigrant individual indicate the complexities they encountered. The reliance on English as a potential alternative language suggests their attempt to bridge the communication gap, albeit with varying degrees of success.

### 3.3. Lack of Training and Skill in Using Interpreter or Translation Tools

Facilitator 2 documented that *“Some students with regional dialects were aware of their potential difficulty in being understood but struggled to adapt their speech to clear Bokmål, the standard Norwegian language”*. Facilitator 2 also noted that *“The students were more concerned with using communication methods they had learned during the topic of motivational communication, than with starting a conversation. For example, the facilitators observed that the students did not want to ask “yes” and “no” questions, because they had been taught that one should try to avoid this. They also forgot to ask about their personality because they were busy telling the clients that they had a duty of confidentiality”*.

All facilitators agreed that the frustration among the students was evident when the clients struggled to comprehend their messages. Facilitator 2 highlighted that “*The students repeated the same words the client did not understand several times without further explanation*”. The frustration became obvious: “*The students were surprised that they got a client who had little knowledge of Norwegian, a communication scenario they had not previously practiced*” (facilitator 4), which led to uncertainty. As a result, uncertainty arose among the students, as expressed by student 10: “*It is difficult to be sure that the information is understood*”.

To address these challenges, facilitator 3 suggested “...*using more active body language and making direct eye contact*” to enhance client comprehension and cooperation. Additionally, several students emphasized the importance of having interpreters or translation tools available for healthcare workers to ensure effective communication. The formal or informal nature of translation was deemed less significant by many participants.

### 3.4. Lack of Confidence in Using Online Translation Services

An important observation during the simulation was the limited confidence displayed by the students towards online or phone-based translation services. The students were also reluctant to use Google Translate. After the second session, facilitator 3 wrote that “*The students explained that they did not use Google translate because it was so bad. When they used it in the second review, several were surprised at how well it worked”*. Facilitator 4 wrote that *“One reason why Google translate was not used was that it would distance them from the client. The result of not using translation tools was that it became completely quiet in the room”*.

Another significant observation highlighted by all facilitators was the persistent use of English by the students when communicating with the individuals, despite discovering that they possessed a similar proficiency in English as they did in Norwegian. All facilitators emphasized that the students began to explain their questions instead of allowing time to respond—thus further complicating the questions for the individuals.

As student 11 stated, “*It may be wise to have tools for intercultural communication*”. Facing challenges other than language difficulties helps to recognize the need to address a broader range of challenges. Another student expressed the view that “*Intercultural communication is so individual; one cannot prepare for all situations*” (student 12). Additionally, several students also acknowledged their oversight, stating that “*We forgot that a large part of the population in Norway are not Norwegians*” (student 15) and *“You should not take it for granted that the person speaks Norwegian”* (student 16). These insights collectively highlight the students’ awareness of the complexities and nuances involved in intercultural communication within a diverse society.

### 3.5. Simulation as a Didactic Approach

As anticipated, our findings revealed a consistent improvement in proficiency among all participants during the second simulation. Facilitator 2 noted that *“The students increased their knowledge and skills in communication with clients with a low language level in Norwegian and all the students performed a better second interview”*.

Even though information was given before the first simulation, it was first comprehended and implemented during the second simulation. When the students were asked during the first debriefing about their experience, the students were surprised about their own reactions and challenges. A student in team 1 stated that *“This should be offered to everyone. Simulation is important. I have learned a lot that I want to take further”*. A student from team 2 reflected on different aspects, stating, “*I realize that, if I am going to do a motivational interview another time, I should practice more in advance. I also need to become more adept at explaining difficult words or using a lighter vocabulary. Debrief gave [me a] good [basis for] reflection”*.

One overarching lesson learned by all students was the recognition that *“Language can be a daunting challenge when you do not understand each other. I experienced the situation as uncomfortable during the first exercise” (student team 4) and that “Being thrown into it has been instructive”* (student team 5). 

Simulation training was regarded as a suitable method to enhance communication skills in the absence of a shared language. One student in team 3 described it as follows:


*“Multicultural communication is very individual, so you cannot be prepared for every situation, but we grow with the challenges we did not expect. There is a lot of learning from having to step out of your comfort zone, as one does during simulation training.”*


## 4. Discussion

In this study, we aimed to investigate the feasibility of employing simulations as an educational tool to equip public health students with the necessary skills to navigate multicultural settings effectively and enhance health outcomes. 

### 4.1. Communication

Our results indicate that effective communication—to understand and be understood—not only meets the needs of clients, but also the needs of the healthcare workers. Furthermore, it is crucial for public health students to recognize the impact of culture on the interactions between healthcare providers and clients. During the simulations, the student teams practiced on immigrants who pretended to be bewildered and intimidated in a complex healthcare system. A study with 36 nursing students for the measurement of critical thinking skills and simulation-based performance found that the students had difficulty meeting expectations in the clinical scenarios [26]. What was unclear, however, is whether simulations are suitable for training public health providers in meetings with immigrants. Another study suggests the necessity for novel strategies to enhance medical training, enabling healthcare workers to navigate culturally complex encounters with clients and their families [27]. Quail et al. (2016) also indicated that participants from health education programs self-report higher communication skills, knowledge, and trust following the implementation of standardized, virtual client-based communication models [28]. The results are consistent with our study results where the participants managed to conquer the communication challenges, regardless of how difficult the conversations between the clients and the students initially were.

### 4.2. Mastery When the Students Repeated the Simulation after Debriefing

Simulations must be designed, executed, and followed with a debriefing to achieve positive learning [13]. In our study, communication mastery was enhanced when the students repeated the simulation after debriefing. Debriefing played a crucial role in the students’ learning journey as they repeated the simulation. After the initial round, it became evident to the students that their understanding of communication with clients from minority backgrounds was not as comprehensive as they had initially believed. During the debrief, the students asked several questions about how they could have performed differently and whether the tutors could exemplify their statements. With a better grasp of the concepts discussed during the debriefing, the students entered the second round feeling more prepared. Remarkably, all students demonstrated notable improvement in their performance. They discovered that effectively communicating with clients using concise and precise sentences presented a unique challenge, requiring dedicated practice to master.

### 4.3. Simulation: Confronting Unexpected Barriers

Simulations have already been used to teach meetings with immigrants where communication is a challenge [29] and where interpreters are not present. Health faculties must create learning environments that help students to become more culturally aware of themselves and others [30]. As our results show, a simulation scenario creates a safe learning environment where students can experience otherness and learn how to act within the given framework. Lack of a common language will not be the only challenge healthcare workers will be confronted with. 

Western healthcare is increasingly serving a multi-ethnic society. Yet language barriers still make it difficult to provide adequate health care to persons with limited ability to communicate in the local language, resulting in significant and costly health disparities. To overcome these difficulties, the healthcare worker must achieve clear and effective communication using all available tools and resources [31,32].

That many immigrants have greater health problems than the native population is well known, both in Norway and in the rest of Europe. Recent research reveals a little-known connection: immigrants who have lived in Norway for a long time increase their likelihood of having health problems if they do not learn the language well [30]. Recent data also show that immigrants have a higher probability than the rest of the population of contracting various ailments such as diabetes, cardiovascular disease, mental health problems, and back and neck pains. This probability is independent of length of residence, knowledge of the Norwegian language, and education [30].

The Norwegian Health and Care Services Act (2011) specifically requires that “health professionals at all levels must have knowledge of the prevalence of various immigrant groups’ illnesses and of cultural challenges related to ensuring immigrants can access an equal health and care service” [31]. However, this can only happen if clients are informed about health services, their rights, and of their own illnesses. To ensure that all clients receive equal health services, it is important that healthcare workers understand the importance of clients’ ethnicities and cultures, and the benefits of the health services offered [32]. By undergoing simulations with role players, healthcare workers will not only learn to communicate with immigrants but also appreciate cultural otherness and become aware of their own attitudes towards people who do not speak their language [33].

A survey on immigrants’ living conditions in Norway commissioned by the Directorate of Health (2016) showed that the relationship between migration and health is complex. Immigrants experienced a decline in health at a younger age than their Norwegian counterparts. Higher age had a stronger association with poor health for immigrants than for the rest of the population. In other words, immigrants tended to experience a deterioration in health at a younger age than the rest of the population. Health also deteriorated with a longer duration of stay in Norway [34].

### 4.4. Strengths and Weaknesses

One of the strengths of this study is that the actors in the simulation scenarios were not mannequins, but employees from Kristiania University College’s School of Health Sciences in the Faculty of Health who had considerable experience in education and research. They brought to the study a user perspective, and their feedback helped verify the data collected from both students and facilitators. Second, an extensive systematic search of the relevant literature allowed us to gain valuable insights and shape the direction of our study. However, unlike a systematic review, we did not employ a flowchart for article selection or create a summary table of the analyzed articles.

One of the weaknesses is that one of the facilitators was the head of the study, thus opening the door to observer bias. The facilitator would have had the opportunity, if not the incentive, to produce favorable results, and this may have tainted the study’s value. 

Another weakness is that the study was performed with a convenience sample of female students who were taking a public health course. This prevented us from investigating any possible effect of gender in communication. The findings should be interpreted cautiously and may not be applicable to male students. This study also includes all the limitations of the participant observer study, particularly types of response bias. For further studies, we recommend including both genders in the simulation and testing the curriculum’s learning outcomes before and after simulation training.

## 5. Conclusions

This study highlights the crucial role of simulations for effective communication in meeting the needs of individuals with immigrant backgrounds and healthcare professionals. The results demonstrate that both students and facilitators gained valuable insights and agreed that simulations in an intercultural and interlingual setting were a suitable approach for improving communication skills. The participants’ ability to navigate the complexities of intercultural communication was enhanced through the simulation-based training and subsequent debriefing sessions. The findings underscore the significance of debriefing as an essential component of the learning process. Debriefing facilitated reflection, self-assessment, and the acquisition of new knowledge and skills, enabling students to identify areas for improvement and explore alternative strategies for better communication.

Furthermore, this study highlights the need for healthcare education programs to integrate cultural competence simulation training and broaden the scope of medical training to address culturally challenging encounters. By acknowledging and addressing these communication challenges, healthcare professionals can deliver more patient-centered care and enhance health outcomes for individuals regardless of their cultural or linguistic backgrounds.

## Data Availability

The data presents in this study are available on request from the corresponding author.

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
