# Peer review of "Using Simulations to Help Public Health Students Overcome Language Barriers for Better Health Outcomes"

_ijerph, 2023, doi:10.3390/ijerph20136259_

Round 1

Reviewer 1 Report

I congratulate the authors for articulating your study in a very clear and easy-to-grasp language. The paper indeed is very well written, however, I do have some concerns and suggestions for improvement:

. Introduction: The second para in the Introduction uses the term, "foreign client'. The same para concludes with the question: " How healthcare professionals treat clients from different ethnic backgrounds will determine whether the service is indeed equal." The term 'clients' and 'foreign' are contradictory to each other. I think the term 'foreign' is highly inappropriate given the people this study is referring to are legitimate migrants and many might be citizens of Norway. However, if you do not wish to remove the term 'foreign', then you may say that many healthcare professionals, esp those who speak Norwegian still consider citizens from different ethnic backgrounds as 'foreign', hence the need to improve communication skills is not only an issue of competence but of attitude as well. 

Methods and data collection: It is not clear what exactly was the simulation? Can that be elaborated/described in stepwise way? Also Observation is not described as the data collection tool. 

Results: The results section is very thin and does not generate any interest. It is not clear how did the two themes emerged? In fact 'communication' and 'simulation as  didactic approach' are not themes that result from the data but categories you wished to explore in the first place.  You may wish to revisit the data analysis to unpack what did you find rather than what were you exploring. 

Reviewer 2 Report

The effectiveness of simulation has already been extensively supported by literature in various fields. However, the manuscript's scientific structure and research methodology are not well-designed. It is necessary for the researchers to revise the manuscript comprehensively. Furthermore, it was challenging for me to understand the research questions used to derive the results. The research questions presented in the [Introduction] section do not align with the research findings.

For instance, the research question stated: "Can simulation as a didactic approach reduce the language and communication challenges that our students and minority language clients experience, and promote health workers’ interaction skills?" Therefore, it is unclear whether the researcher intends to answer this research question with a simple "yes" or "no." It is advisable to reformulate the research question to adopt a more exploratory and qualitative research approach.

In the [Method] section, the researcher mentioned a mixed methods design, but there are various types of mixed methods designs. It would be helpful to specify the exact type of mixed methods design employed in this study.

The researcher stated that a structured literature search was conducted in the [Method] section. However, it does not appear to be a systematic literature review approach. It would be beneficial for the researcher to clarify why the ORIA database was selected among several databases and provide a rationale for the set limitations, such as "The search was limited to the Norwegian, Swedish, Danish, English, and German languages for the years 2012 to 2022." When setting such limitations, they should be based on relevant literature. Additionally, the researcher should include a flowchart demonstrating the selection process of the last 10 articles searched and summarize the analyzed articles in a table.

If phenomenological analysis was employed in the [Method] section, it is essential to specify whose methodology was followed. The references listed in the literature do not indicate the publication year. If possible, it would be helpful to provide specific references to accurate phenomenological scholars. The researcher should describe the data analysis method in detail to enable other readers to replicate and implement this research approach.

In the [Results] section, it would enhance the manuscript's readability and reader understanding to present the results using tables or figures.

Regarding the statement in the [Method] section - "The study and its methodological approach were presented to the Norwegian Center for Research Data. Due to the outline of the study, no further approvals were necessary. All students were informed about the study before they signed their consent forms" - stating that "no further approvals were necessary" is the researcher's subjective opinion. If the study did not require approval, it would be appropriate to provide an exemption approval number.

In the [Conclusion] section, it is advised to avoid such descriptions and instead focus on the impact of the research findings on the clinical field. For example, instead of the provided description, the researcher should emphasize the implications of the research results in the clinical setting.

Round 2

Reviewer 1 Report

It's good to see that suggested changes have been taken on board and that has certainly strengthened this manuscript.